# Multi-Scale Feature Fusion Network with Symmetric Attention for Land Cover Classification Using SAR and Optical Images

Dongdong Xu [1,*], Zheng Li [1,2], Hao Feng [1,2], Fanlu Wu [1] and Yongcheng Wang [1]

1   Changchun Institute of Optics, Fine Mechanics and Physics, Chinese Academy of Sciences, Changchun 130033, China; lizheng20@mails.ucas.ac.cn (Z.L.); fenghao21@mails.ucas.ac.cn (H.F.); flwu@ciomp.ac.cn (F.W.); wangyc@ciomp.ac.cn (Y.W.)
2   University of Chinese Academy of Sciences, Beijing 100049, China
*   Correspondence: xudongdong@ciomp.ac.cn

**Abstract:** The complementary characteristics of SAR and optical images are beneficial in improving the accuracy of land cover classification. Deep learning-based models have achieved some notable results. However, how to effectively extract and fuse the unique features of multi-modal images for pixel-level classification remains challenging. In this article, a two-branch supervised semantic segmentation framework without any pretrained backbone is proposed. Specifically, a novel symmetric attention module is designed with improved strip pooling. The multiple long receptive fields can better perceive irregular objects and obtain more anisotropic contextual information. Meanwhile, to solve the semantic absence and inconsistency of different modalities, we construct a multi-scale fusion module, which is composed of atrous spatial pyramid pooling, varisized convolutions and skip connections. A joint loss function is introduced to constrain the backpropagation and reduce the impact of class imbalance. Validation experiments were implemented on the DFC2020 and WHU-OPT-SAR datasets. The proposed model achieved the best quantitative values on the metrics of OA, Kappa and mIoU, and its class accuracy was also excellent. It is worth mentioning that the number of parameters and the computational complexity of the method are relatively low. The adaptability of the model was verified on RGB–thermal segmentation task.

**Keywords:** land cover classification; SAR and optical images; attention mechanism; multi-scale feature fusion; semantic segmentation

## 1. Introduction

Semantic segmentation refers to pixel-level annotations and different types of objects can be distinguished in segmented maps. It is a more refined task than classification and detection, and has been widely developed in assisted driving, geological detection and medical image analysis, among other scenarios. In particular, in Earth observation (EO) missions, land use and land cover (LULC) classification has become a key link in remote sensing (RS) data interpretation. Such classification results are already used for crop monitoring, urban development planning, disaster response and other tasks [1,2]. However, most of the common segmentation methods are based on unimodal RS images [3], which are insufficient for complex scene representation [4]. Spectral confusion or noise interference often affects the accuracy of classification. With the continuous development of sensors and imaging techniques, it becomes slightly easier to acquire multi-modal remote RS images of the same region simultaneously [5]. More comprehensive information about land cover can be acquired, further meeting the needs of advanced vision tasks. Optical images such as multi-spectral and hyperspectral images are still the primary data used for remote sensing classification [6]. The spatial resolution of these images is high, and more details of ground objects can be preserved. However, the imaging process is often disturbed by weather factors, especially frequent occlusion by clouds and fog. This is the drawback

of catoptric imaging. In contrast, radar devices such as synthetic aperture radar (SAR) generate images by continuously transmitting microwaves and using scattered echoes. They are not easily disturbed and can almost work in all-day and all-weather conditions [7,8]. Therefore, SAR images can provide structural and electromagnetic scattering information but suffer from severe speckle noise, resulting in lower resolution [9]. It is easy to see that optical and SAR images are obviously complementary. Some objects that are difficult to recognize in a unimodal image might be clearly identified in another modal image. Therefore, the joint use of multi-modal image data is beneficial in improving the accuracy of land cover classification [8,10].

The joint application of optical and SAR images for semantic segmentation has been of great interest. The multi-modal classification methods can be roughly grouped into two categories, which are conventional machine learning and deep learning methods [2,11]. In the first category, support vector machines (SVMs) [12], conditional random fields (CRFs) [13], random forest (RF) [14], K-nearest neighbors (KNNs) [15] and other nonparametric approaches have been applied to classification tasks [16]. The above methods have achieved some classification accuracy. However, due to their weak feature extraction ability and insufficient high-dimensional information representation, they cannot obtain better classification results. Recently, deep neural networks with powerful feature extractors have shown great advantages in multi-modal classification tasks [17–19]. These methods can be subdivided according to the fusion level of the inherent information. Pixel-level fusion exists in early networks. Original pixels are fed to the multi-layer perceptron to aggregate the predictions [7]. The contextual information and the correlation between pixels are ignored. Decision-level fusion is applied in the late stage and depends on the results of several methods. The drawback is that the multi-dimensional features from different modalities are not considered [2]. At present, intermediate feature-level fusion, which focuses on the extraction and transformation of semantic features, is a research hotspot. The most dominant methods to effectively obtain and fuse the multi-modal information from optical and SAR images are the two-branch end-to-end segmentation models without weight-sharing [2,3,5,7,9,20]. These supervised methods have received the most attention and are constantly improving. The attention-based MCAM [2] module, the SACSM [3] module, the SaC [7] module and the SEPP [9] module have been used for salient feature extraction. Employed fusion strategies include the gate methods of GHFM [5] and CRGs [9], the cross-fusion method reported in [1], etc. Other optimizations with respect to pre-processing and loss functions are also considered. The ultimate goal is to realize the representation and complementary utilization of high-dimensional semantic features of different modalities. The supervised methods can achieve high accuracy, but they rely on registered images with semantic labels when training and testing. In reality, we may not obtain usable optical images immediately or there may be only a single unimodal image available at a time. This could lead to a rapid degradation of multi-modal classification performance. To this end, semantic knowledge distillation has been introduced for knowledge transfer and aggregation [21,22]. To address the scarcity of labeled data, some researchers have adopted self-supervised learning to realize joint segmentation with SAR and multi-spectral images [23–25]. In addition, with the further innovation of deep learning frameworks, graph convolution networks [26] and transformers [27] are gradually emerging in LULC classification tasks.

As described above, although the supervised two-branch models have some constraints, they are of important research significance for the joint classification of multi-modal SAR and optical images. After analysis and comparison, we think that there are still several challenges to be overcome. First, multi-modal semantic features are not effectively extracted. Attention modules [2,3,7,9] with square convolution kernels are defective for the representation of irregular objects. Anisotropic contextual information should be further integrated. Secondly, the fusion strategies for multi-scale features need to be improved. Existing methods [1,5,9] usually focus on high-level features, while low-level features and other complementary information are ignored. The semantic inconsistency of different modalities cannot be mitigated. Thirdly, multi-modal registered datasets with SAR and optical images for multi-class segmentation are extremely scarce [2,24,25]. The generaliza-

tion and adaptability of the models have to be considered. Finally, the network structures have become gradually complicated to obtain higher classification accuracy. It is worth thinking about how to ensure the performance of the models with as few parameters and computation as possible.

The work of this paper is aimed at the situation and existing difficulties, and the main contributions are summarized below.

1. We propose a multi-modal segmentation model for the classification of optical and SAR images. It is an end-to-end network (SAMFNet) based on a multi-layer symmetric attention module and multi-scale feature fusion module. There are no other pretrained backbones in the framework.

2. A novel symmetric attention module is constructed with strip pooling. Multiple long receptive fields help to obtain more complementary and contextual information from the two branches. Atrous spatial pyramid pooling, varisized convolutions and skip connections are tactfully combined to fuse the multi-scale and multi-level semantic features.

3. The proposed model achieves the best numerical and visual results on two available datasets. The applicability of the model is proven on another RGB–thermal segmentation task. The designed network is relatively lightweight, and the computational costs and parameters are low, considering its classification accuracy.

## 2. Materials and Methods

### 2.1. Data Preparations

The SEN12MS is a curated dataset composed of dual-polarimetric SAR, multi-spectral images and MODIS (Moderate-Resolution Imaging Spectroradiometer)-derived land cover maps [28]. The first two are from Sentinel-1 and Sentinel-2. Multi-modal data were collected from regions of interest around the world with four seed values. There are 180,662 image patches in total with a size of $256 \times 256$. The ground sampling distance (GSD) of the original data can reach 10 m, but the land cover maps with labeled classes are at a lower resolution of 500 m. They are relatively crude for specific classification or detection tasks. In the 2020 IEEE GRSS Data Fusion Contest [29], the source images are the same as SEN12MS, and some high-resolution (10 m) labels were semi-manually generated for validation based on the original MODIS maps. As a result, 6114 image patches with high-resolution labels were obtained to construct the DFC2020 dataset. Figure 1 shows visual examples of the DFC2020 data.

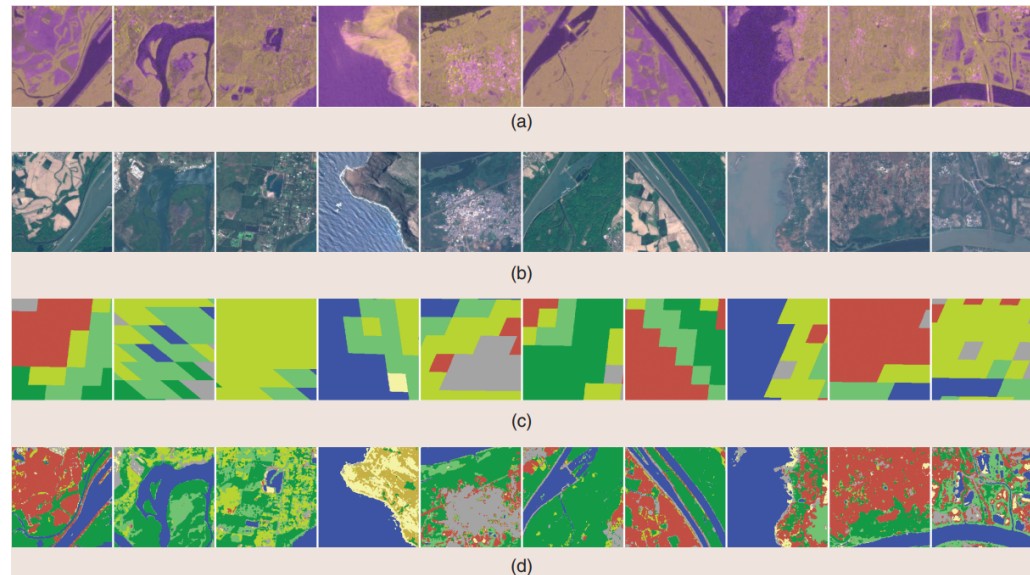

**Figure 1.** Source images and labels of DFC2020 [29]. (**a**) SAR images of Sentinel-1. (**b**) Optical images of Sentinel-2. (**c**) Low-resolution semantic labels. (**d**) High-resolution semantic labels.

As to WHU-OPT-SAR [2], there are 100 pairs of SAR and optical images from GF-3 and GF-1 with a size of 5556 × 3704. The imaging areas are all in Hubei Province, and the resolutions of source images and labels were unified to 5m. Figure 2 shows two examples of WHU-OPT-SAR data.

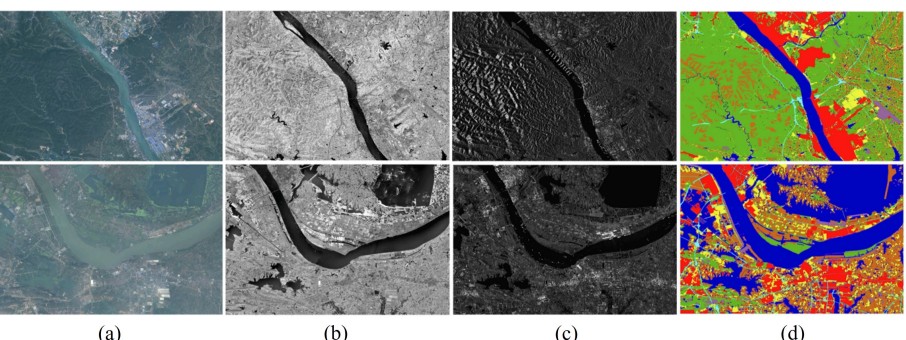

(a)                         (b)                         (c)                         (d)

**Figure 2.** Source images and labels of WHU-OPT-SAR [2]. (**a**) RGB images of GF-1. (**b**) NIR images of GF-1. (**c**) SAR images of GF-3. (**d**) Semantic labels.

In order to train the model on these two datasets under a unified framework, the pre-processing of the data should be carried out in advance. Firstly, the number of images in WHU-OPT-SAR is small, but the image size is quite large. Therefore, augmentation was performed by cropping with a proper stride. Then, tens of thousands of patches with a size of 256 × 256 were obtained. Secondly, the number of channels of the multi-modal inputs needs to be consistent. Single-channel gray SAR images and four-channel (RGB and NIR) optical images were adopted. Since the multi-spectral images of DFC2020 have 13 bands, B4, B3, B2 and B8 were chosen to combine the corresponding RGBN inputs. Furthermore, flipping and scaling at multiple scales were also executed when importing the training data to improve the generalization capability of the model. Other details are presented in the subsequent experiments.

### 2.2. Attention Mechanisms for SAR and Optical Image Classification

In complex scenes, semantic segmentation networks usually face challenges such as information redundancy and poor relevance. In order to overcome these problems, over the years, researchers have introduced several attention mechanisms to the semantic segmentation models. They are able to extract more salient features adaptively, so the performances of diverse networks are effectively promoted [30–33]. For the joint classification of SAR and optical images, methods with attention mechanisms can be mainly classified as two types. Some methods are built with channel attention [3], spatial attention [34] or their combined block modules [7], such as CBAM [35]. By introducing these mechanisms, the models are able to learn the importance of channels and regions of the source images automatically. They pay more attention to the target itself and ignore the contents that are detrimental to the classification task. The multi-scale information is also focused [36,37]. Other methods are based on self-attention [2,9,16]. The essence of the self-attention mechanism is to perform a mapping from a query to a sequence of key-value pairs. The correlations between the matrices are adequately calculated. Its advantage lies in the ability to establish global dependencies and capture internal correlations of features.

In fact, increasing the receptive field is the main purpose of the attention-based semantic segmentation methods. Self-attention can establish long-range dependencies, but such methods require large memory for complex matrix calculations. Dilated convolution [38] and global/pyramid pooling [39,40] have also been used to improve the receptive field, but they are both confined to the square convolution kernels [41]. To this end, strip pooling (SP) with two-cross long kernels has been proposed for salient feature extraction, which can be seen as an effective attention mechanism [41–43]. Encoders with strip pooling are able to probe the input feature maps through long windows so that objects with irregular

shapes are easy to process and more anisotropic contextual information in complex scenes can be obtained. In view of the above advantages, this paper introduces the use of pooling into multi-modal semantic segmentation to solve the problems of detail errors and class confusion in land cover classification.

### 2.3. Proposed SAMFNet

#### 2.3.1. Framework of the SAMFNet

The proposed SAMFNet (Figure 3) is a concise end-to-end model, and the two inputs are single-channel SAR images and four-channel optical images. The symmetric attention extraction module (SAEM) is embedded at the medial axis of the network. It can extract and supplement distinguishing features to each convolutional group of the two branches for subsequent calculations. With the deepening of the network layers, features obtained by the attention mechanism are gradually refined. More contextual information can be gathered for feature encoding. Atrous spatial pyramid pooling (ASPP) [38] and convolutions with varisized kernel sizes are combined as the multi-scale semantic extraction module (MSEM) to obtain more high-level features. To complement more low-level features, skip connections are added to realize information transmission from shallow layers to deep layers. Then, the multi-level semantic features are concatenated and upsampled for decoding. The encoding and decoding processes of the proposed structure are relatively simple, but the performance of the network is pretty good. Meanwhile, the SAEM and MSEM can be replaced flexibly if other better feature extraction modules are explored.

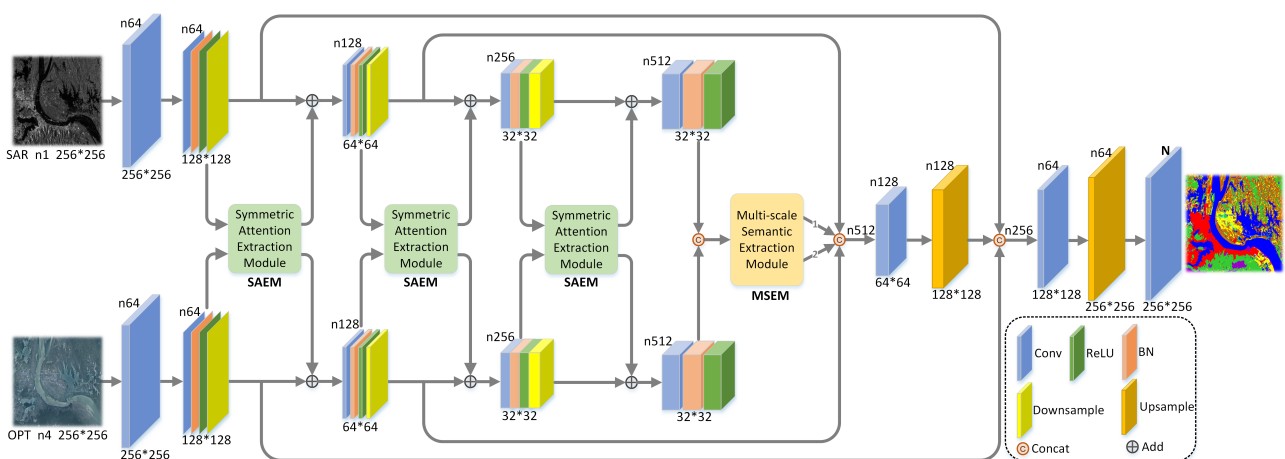

**Figure 3.** The framework of the proposed SAMFNet. n represents the number of channels.

#### 2.3.2. Symmetric Attention Extraction Module

The SAEM designed in this paper is inspired by the multi-modal transfer module (MMTM) [44]. The reason it is called symmetric attention is that the inputs of this module are the multi-modal features from the two branches. Then, the obtained salient features are fed back to the original branch. Figure 4 shows the detailed form of the module. The transformation process is annotated with symbols.

The leftmost part of the figure shows preliminary feature convolution with $1 \times 1$ kernels, halving the number of the concatenated channels. $F_{in1}$ and $F_{in2}$ represent the inputs from the two branches. $F_B$ is the basic feature map, which is actually a four-dimensional tensor. A single-channel and two-dimensional map with a size of $H \times W$ is taken as an example for explanation of the pooling process.

$$F_B = Conv(F_{in1}, F_{in2}) \tag{1}$$

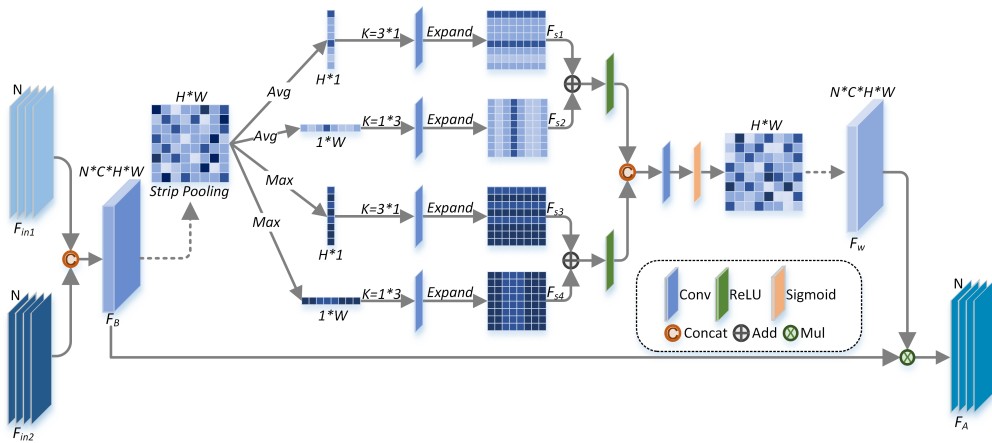

**Figure 4.** The structure of the symmetric attention extraction module.

It can be seen that four strip pooling branches (two average pooling and two max pooling) were designed for feature representation. The map is compressed into a single row or column after pooling. More global information is obtained through these long receptive fields. After each pooling, strip convolution with a specific kernel is performed for further feature transformation. Then, the single row or column features are expanded to a size of $H \times W$. $F_{s1}$, $F_{s2}$, $F_{s3}$ and $F_{s4}$ are acquired as the outputs of strip pooling.

$$F_s = Exp(Conv(Pool(F_B)))$$ (2)

The $F_{s1}$ and $F_{s2}$ are fused, then activated to obtain the final feature maps after average strip pooling. Similarly, the $F_{s3}$ and $F_{s4}$ are combined to obtain the feature maps after max strip pooling. Then, they are concatenated, and the feature weights ($F_w$) are acquired after convolution and nonlinear computation. At last, element-wise multiplication operations between $F_B$ and $F_w$ are implemented to calculate the ultimate highlighted feature ($F_A$) maps after the attention extraction module.

$$F_w = Sigmoid(Conv(ReLU(F_{s1} + F_{s2}), ReLU(F_{s3} + F_{s4})))$$ (3)

$$F_A = F_B \otimes F_w$$ (4)

The unique advantage of strip pooling is that the long-range dependencies can be established easily. Average and max pooling with row and column transforms are introduced together, so land objects with different shapes and scales can be portrayed more accurately. Simultaneously, the particular strip forms can also remove unnecessary connections between feature maps, which greatly reduces the computational complexity compared to other attention-based algorithms.

### 2.3.3. Multi-Scale Semantic Extraction Module

After multiple groups of convolution transformation and salient feature extraction, high-level features of SAR and optical branches can be built. In fact, multi-scale information is essential for accurate semantic segmentation and other computer vision tasks. Many researchers have focused on this topic. Figure 5 shows the specific structure of the designed MSEM module.

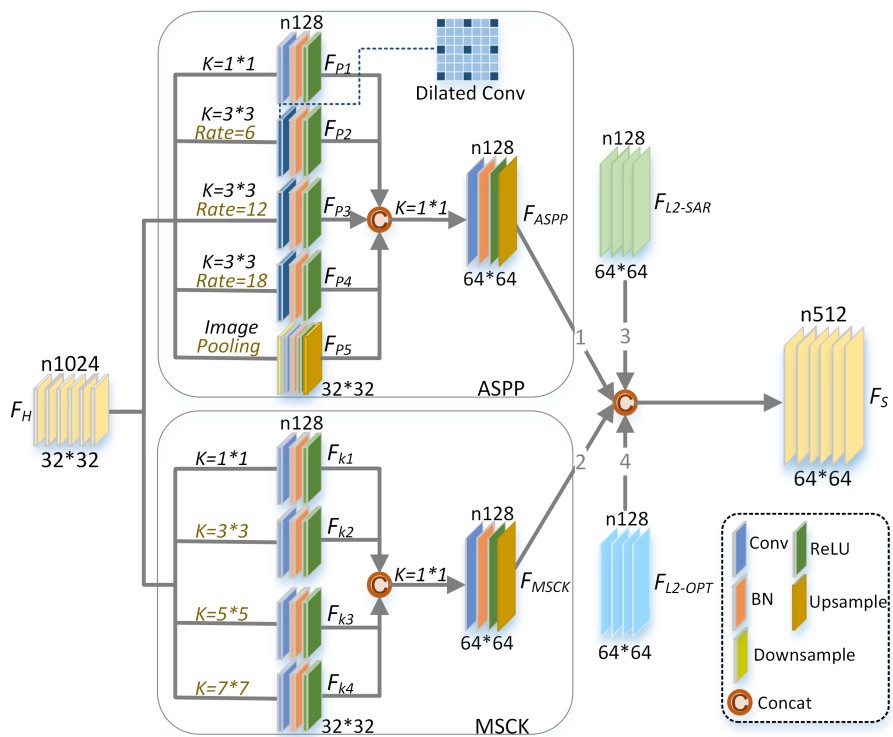

**Figure 5.** The structure of the multi-scale semantic extraction module.

In the figure above, $F_H$ represents the concatenated result of the high-level features from the two branches. ASPP [38] and its variants have been widely adopted in segmentation tasks. They employ several dilated convolutions with different sampling rates to obtain multi-scale information. Such changes allow the network to utilize larger receptive fields without using regular pooling. They can also reduce information loss. The receptive fields are adjusted when setting different dilation rates, and corresponding multi-scale features are easily acquired. The rate group $[6, 12, 18]$ was used in this MSEM module. $F_{P1}$ represents the features calculated by the normal convolution group (Conv + BN + ReLU), while $F_{P2}$, $F_{P3}$ and $F_{P4}$ are obtained by the dilated convolution group. $F_{P5}$ is acquires by the pooling group and upsampling. Then, the feature maps of the five branches are concatenated together. The following convolution group is used to reduce the number of channels to the single branch. It should be noted that we performed an interpolation behind the convolution to maintain consistency with the size of low-level features from shallow layers.

The bottom half of the figure is called the multi-scale convolution kernel (MSCK) module. In the four branches, four different convolution kernels are used to extract specific features. $F_{k1}$, $F_{k2}$, $F_{k3}$ and $F_{k4}$ are constructed with increasing receptive fields. This helps to capture richer multi-scale information and can solve the problem of features of diverse ground objects not being distinctly distinguished in complex scenes. The convolution and interpolation operations are also executed to obtain the $F_{MSCK}$, which has the same tensor form as ASPP. At the end of the module, $F_{ASPP}$ and $F_{MSCK}$ represent the high-level features. $F_{L2-SAR}$ and $F_{L2-OPT}$ from the second convolutional group of multi-modal branches represent the low-level features. Then, they are combined in sequence to build the final multi-scale semantic feature maps ($F_S$). So far, we have completed the feature encoding process of SAR and optical images.

### 2.3.4. Decoding Process

Compared with feature encoding, the decoding process of semantic segmentation is relatively simple. The common methods use multiple groups of convolutions and interpolations to restore the features to the size of original inputs.

In Figure 6, $F_{UP1}$ is acquired directly through the first pair of convolution and interpolation transformation. In the proposed method, in order to better realize feature mapping between the multi-modal inputs and the output, the low-level features ($F_{L1-SAR}$ and $F_{L1-OPT}$) after the first convolutional group of dual branches are added through skip connections. Therefore, more foundational information can be utilized, which is beneficial to improve the classification accuracy. $F_{UP2}$ represents the results of the second interpolation. $F_{Map}$ is achieved after the final convolution operation. $N$ represents the total number of classes of ground objects. Finally, $F_{Map}$ is used to facilitate the calculation of the confusion matrix.

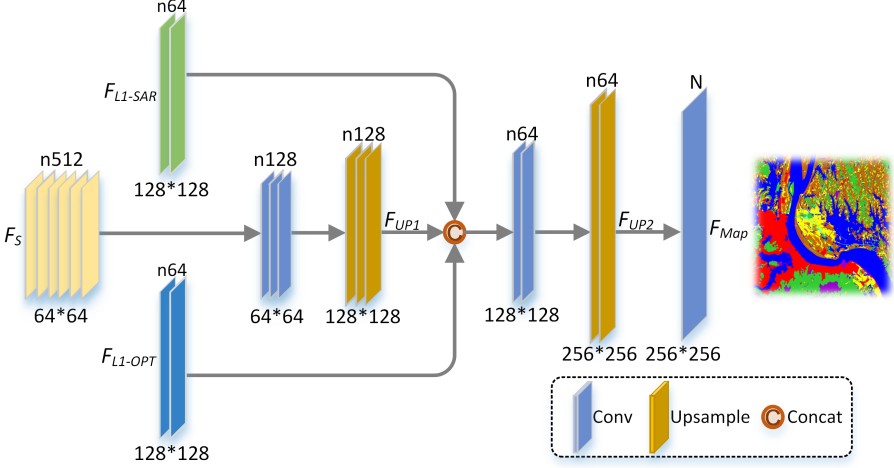

**Figure 6.** The structure of the decoder.

### 2.3.5. Joint Loss Function

In semantic segmentation tasks, pixel-level classification needs to be achieved, so the corresponding pixel-level loss functions are used to constrain network training. However, a single loss function usually cannot describe the segmentation result objectively and comprehensively. The introduction of multiple functions to form a joint loss function [7,15,21] has been proven to be conducive to obtaining better numerical and graphic results. In this article, fully considering the multi-modal characteristics of the input images and the complexity of the classification scenes, the following three loss functions are combined.

$$L = L_{CE} + L_{Focal} + L_{SIoU} \tag{5}$$

where $L_{CE}$ represents the cross-entropy loss, which has been widely adopted in image segmentation and other classification tasks. $L_{CE}$ is defined as follows:

$$L_{CE} = -\frac{1}{N} \sum_{n=1}^{N} y_n log(p_n) \tag{6}$$

where $y_n$ and $p_n$ denote the label and the predicted output of image $n$, respectively. $N$ is the total number of inputs for computation. The definitions of the symbols in the following formulas are the same. The loss focuses on pixel-level information and inevitably ignores the spatial consistency between prediction regions. This may lead to scattered and discontinuous segmentation regions when the amount of different samples is unbalanced. Some researchers proposed using the focal loss ($L_{Focal}$) [45] to address this class imbalance problem. $L_{Focal}$ is defined as follows:

$$L_{Focal} = -\frac{1}{N}\sum_{n=1}^{N} \alpha(1 - p_n e^{-y_n})^{\gamma} y_n log(p_n) \tag{7}$$

where $\alpha$ denotes the conditioning weight of the positive samples. The exponent $\gamma$ is the decay factor of $L_{CE}$. These two hyperparameters are adjusted to balance the positive and negative samples. Moreover, the IoU loss [46] is also taken into account. IoU is the intersection over union. It can be seen as an acknowledged metric for evaluating the effectiveness of object detection and segmentation algorithms. The IoU loss is able to better constrain the similarity between the segmentation results and the true segmentation labels. The soft IoU loss ($L_{SIoU}$) additionally adds softmax to the predicted output for smoothing. $L_{SIoU}$ is defined as follows:

$$L_{SIoU} = -\frac{1}{C}\sum_{c=1}^{C} \frac{\sum_{n=1}^{N} y_n p_n}{\sum_{n=1}^{N}(y_n + p_n - y_n p_n)} \tag{8}$$

where C is the total number of classes. It has been proven that the joint loss function can balance the various optimization objectives and achieve better segmentation results.

## 3. Results

The experiments are presented in detail in the following subsections. More specific analysis of the images and numerical results are provided.

### 3.1. Experimental Settings

The experiments reported in the article were all performed on a server with two RTX3090, and the GPU memory was 24 GB. Pytorch (1.13) on an Ubuntu (18.04) system was used to build the network framework. The Adam optimizer was adopted for parameter updating, and the weight decay was 0.0001. The step size and the gamma of StepLR are 30 and 0.1. The basic learning rate was set to 0.001. To keep the image sizes consistent, all the inputs of the two datasets were cropped to $256 \times 256$. The batch size for training was set to 32. Limited by the capabilities of the server, the image size and batch size were set to $128 \times 128$ and 4 when TAFFN was implemented. A total of 6114 pairs of images of the DFC2020 dataset and 29,400 pairs of the WHU-OPT-SAR dataset were obtained. The ratio of the training set to the testing set was 4:1.

### 3.2. Evaluation Metrics

The segmentation results should be evaluated more accurately, so overall accuracy (OA), Kappa coefficient (Kappa) and mean intersection over union (mIoU) were used for numerical measurements. OA focuses on how well all samples are classified. Kappa is used for consistency checking which can also measure the classification accuracy. mIoU is the average of the ratio of the intersection and union of the true and predicted pairs for each class. It can be seen as a standard metric for semantic segmentation. The formulas of the above three metrics are defined one by one as follows:

$$OA = \frac{\sum_{i=1}^{K} p_{ii}}{\sum_{i=1}^{K}\sum_{j=1}^{K} p_{ij}} \times 100\% \tag{9}$$

$$p_e = \frac{\sum_{i=1}^{K} (\sum_{j=1}^{K} p_{ji} \times \sum_{j=1}^{K} p_{ij})}{(\sum_{i=1}^{K} \sum_{j=1}^{K} p_{ij})^2} \tag{10}$$

$$Kappa = \frac{(OA - p_e)}{(1 - p_e)} \times 100\% \tag{11}$$

$$mIoU = \frac{1}{K} \sum_{i=1}^{K} \frac{p_{ii}}{\sum_{j=1}^{K} p_{ij} + \sum_{j=1}^{K} p_{ji} - p_{ii}} \times 100\% \tag{12}$$

where $K$ is the total number of classes. It is equal to the width of the confusion matrix. $p_{ij}$ is the amount of pixels in class $i$ predicted as class $j$. Higher values of the three metrics indicate better results.

### 3.3. Experimental Analysis

In order to illustrate the feasibility of the proposed method, DeeplabV3+ [47], DenseASPP [48], DANet [30], CCNet [31], TAFFN [34] and MCANet [2] were selected for comparison. Among them, the attention mechanism is not introduced in DeeplabV3+ and DenseASPP. They mainly perform semantic feature transformation through dense connections and ASPP. Different attention methods are separately introduced in the remaining four methods for salient feature extraction. The codes used for validation are mainly provided by the authors or codebase. For the sake of fairness, the data loader, training process and loss function of these methods are consistent with the proposed method.

### 3.3.1. Experiments on DFC2020 Dataset

Comparative experiments were implemented on DFC2020, and the numerical results, including the accuracy of each class, are listed in Table 1. As shown in the last row, the proposed model achieved the best numerical results on OA, Kappa and mIoU among all the compared methods. These three metrics increased by 3.4%, 4.2% and 7.3%, respectively. The reasons for the better results lie in the following aspects. In the stage of feature encoding, the series-wound symmetric attention module on the central axis is a contributing factor. The horizontal and vertical receptive fields are conducive to acquiring and transmitting multi-size contextual information. The gradually refined features are then fed back to each branch for secondary learning. Another important point is the integration of multi-scale features. This improves the semantic understanding of images. Both the enhanced high-level features and the shallow low-level features from the two branches are used to fuse the final output of the encoder. All these lay the foundation for decoding accurate segmentation maps. In a word, the effective extraction and interaction of complementary features are key to improving the numerical metrics. CCNet obtained suboptimal OA and Kappa values. The recurrent criss-cross attention focuses on full-image contextual information, but the mIoU is low. Partly because the number of parameters is large, it is difficult to train a stable network on a smaller dataset. MCANet performed well on all three metrics. The multi-modal cross-attention module and feature fusion module were introduced together to retain more features. DeepLabV3+ and DANet achieved similar results. They benefit from the use of ASPP and dual attention, respectively. The values of DenseASPP were slightly lower. The special structure with dense connections needs more data to optimize the weights. As to TAFFN, due to its high computational complexity, the existing server cannot support training when the images are large. Therefore, both experimental calculations and classification maps are based on the central part of the images (128 × 128). The image size of other methods is 256 × 256. Therefore, the results may be improved if the server has enough capacity. The comparison of parameters and computation is explained in the Section 4.

Meanwhile, the proposed method can also achieve maximum classification accuracy for each land cover target. This is particularly prominent in the classification of shrubland, grassland and barren. These three objects have many interaction areas with other objects (shrubland and forest, grassland and forest, and barren and cropland). They can be segmented more accurately because the multi-modal complementary information and high-level semantic features are all taken into account by the proposed method. From the numerical analysis, we can see that the proposed model has certain advantages.

The source images and classification maps of diverse models are demonstrated in Figure 7. Different land covers are distinguished by colors. The four selected groups of typical images contain all the categories. In the first group, in addition to the large area of forest and grassland in the label image, the meandering water body and its surrounding scattered wetland and cropland are the focuses of segmentation. DeepLabV3+ and DANet can mainly segment the obvious target area, and the degree of fine classification is not enough. In the image of DenseASPP, there are confusions between forest and grassland, wetland and water. The proposed method, MCANet and CCNet achieve better results. In the latter three groups of images, the interactions between different ground objects are more serious. In the second group, the proposed method, CCNet and DenseASPP work well. As to MCANet, large areas of wetland are misclassified as cropland. The superiority of the proposed model is more obvious in the third group. The forest and shrubland are almost mixed into one class in DeepLabV3+, DenseASPP, DANet and MCANet. At the same time, the distribution of barren is not well reflected. In the last group, all kinds of ground objects can be well reflected by MCANet and the proposed method. The accuracies of forest and barren need to be improved in other methods. The four segmentation images of TAFFN can generally reflect different classes of ground objects, but there are some obvious misclassified areas.

**Table 1.** Experimental results on DFC2020 dataset (Optimal values are in bold).

| Models | Class Accuracy | | | | | | | | OA | Kappa | mIoU |
|---|---|---|---|---|---|---|---|---|---|---|---|
| | Forest | Shrubland | Grassland | Wetland | Cropland | Built-Up | Barren | Water | | | |
| DeepLabV3+ | 0.8802 | 0.5803 | 0.6421 | 0.5638 | 0.8410 | 0.8320 | 0.4996 | 0.9890 | 0.8297 | 0.7920 | 0.6106 |
| DenseASPP | 0.8115 | 0.4729 | 0.7517 | 0.5612 | 0.7774 | 0.8911 | 0.4681 | 0.9866 | 0.8109 | 0.7706 | 0.5831 |
| DANet | 0.8856 | 0.5513 | 0.6617 | 0.5593 | 0.8257 | 0.8347 | 0.4367 | 0.9881 | 0.8267 | 0.7882 | 0.6022 |
| CCNet | 0.9159 | 0.5784 | 0.7330 | 0.5985 | 0.8290 | 0.8407 | 0.4398 | 0.9723 | 0.8419 | 0.8070 | 0.5028 |
| TAFFN | 0.9137 | 0.1022 | 0.4898 | 0.3616 | 0.7219 | 0.8258 | 0.0230 | 0.9904 | 0.7491 | 0.6897 | 0.4439 |
| MCANet | 0.8880 | 0.5655 | 0.7372 | 0.5541 | 0.8149 | 0.8699 | 0.3978 | 0.9927 | 0.8374 | 0.8018 | 0.6127 |
| Proposed | **0.9291** | **0.6324** | **0.8247** | **0.6257** | **0.8423** | **0.8922** | **0.5890** | **0.9956** | **0.8763** | **0.8492** | **0.6853** |

We can see that the obtained numerical results are basically consistent with the results of the classification maps. Generally, the proposed method and CCNet achieved the best performances. The segmented maps are closer to semantic labels. On the contrary, the forest and cropland accuracy of DenseASPP, the wetland and barren accuracy of MCANet and the grassland accuracy of DeepLabV3+ and DANet are relatively low. All these drawbacks are well reflected in the final classification maps.

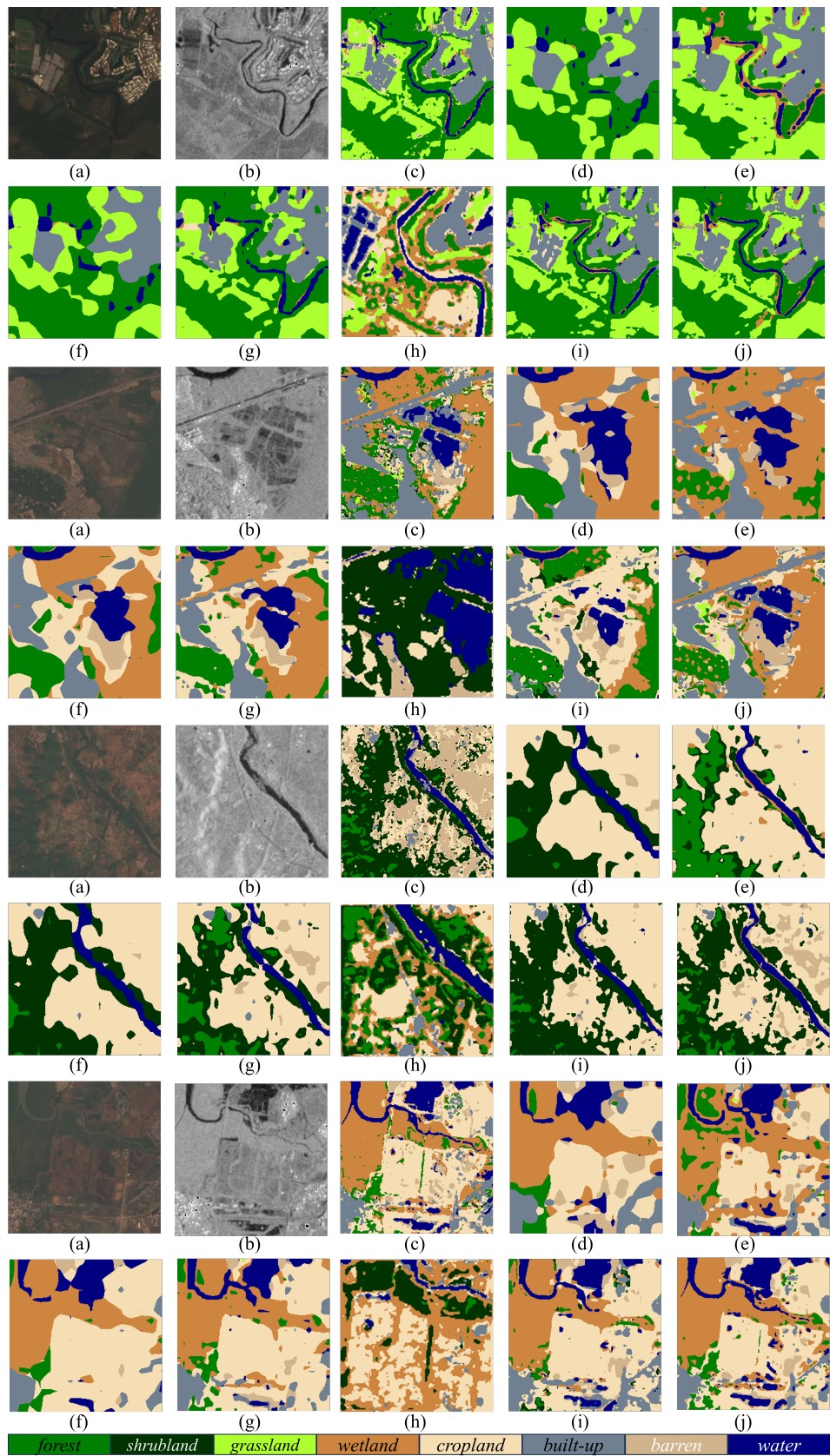

**Figure 7.** Land cover classification maps of different models on DFC2020. (**a**) Optical images. (**b**) SAR images. (**c**) Semantic labels. (**d**) DeepLabV3+. (**e**) DenseASPP. (**f**) DANet. (**g**) CCNet. (**h**) TAFFN (128 × 128). (**i**) MCANet. (**j**) Proposed SAMFNet.

3.3.2. Experiments on WHU-OPT-SAR Dataset

The WHU-OPT-SAR dataset is larger than DFC2020 and is more imbalanced. The relevant numerical results are listed in Table 2. The proposed model also obtained the best results on OA, Kappa and mIoU and achieved excellent performance in class accuracy. Except the OA of DenseASPP and TAFFN, the overall numerical results of these methods are somewhat reduced compared with the first dataset. The increase in data helps to optimize weights and can better leverage the advantages of dense connections and ASPP. Although the number of parameters of TAFFN is small, the computational complexity is very high. More data means more accurate inference. Therefore, the performance of these two methods was improved. CCNet still obtained suboptimal values of OA and Kappa, and the mIoU was also acceptable. DenseASPP obtained a suboptimal mIoU value and exceeded the results of MCANet. The values of DeepLabV3+ and DANet were slightly lower. The results of TAFFN can be improved with the proper setting of parameters and a large amount of computation.

For class accuracy, only the value of villages of the proposed method was slightly lower (−0.0015) than MCANet. The classification effects on water, road and other land use types were more excellent. This is attributed mainly to the efficient extraction and fusion of multi-modal salient features and multi-scale contextual information. These objects are linear, curving or scattered. The improvement of their accuracy is conducive to fine segmentation.

**Table 2.** Experimental results on WHU-OPT-SAR dataset (Optimal values are in bold).

| Models | Class Accuracy | | | | | | | OA | Kappa | mIoU |
|---|---|---|---|---|---|---|---|---|---|---|
| | Farmland | City | Village | Water | Forest | Road | Others | | | |
| DeepLabV3+ | 0.8277 | 0.7636 | 0.6289 | 0.7777 | 0.8988 | 0.5420 | 0.3132 | 0.8180 | 0.7427 | 0.4786 |
| DenseASPP | 0.8290 | 0.7632 | 0.6560 | 0.8088 | 0.8998 | 0.5433 | 0.3307 | 0.8251 | 0.7532 | 0.4912 |
| DANet | 0.8255 | 0.7632 | 0.6246 | 0.7856 | 0.8952 | 0.4970 | 0.2860 | 0.8158 | 0.7397 | 0.4726 |
| CCNet | 0.8370 | 0.7527 | 0.6380 | 0.8175 | 0.8990 | 0.5613 | 0.2913 | 0.8268 | 0.7551 | 0.4908 |
| TAFFN | 0.7951 | 0.7298 | 0.4018 | 0.7146 | 0.8681 | 0.1424 | 0.0005 | 0.7622 | 0.6598 | 0.3602 |
| MCANet | 0.8269 | 0.7367 | **0.6713** | 0.8039 | 0.8990 | 0.5423 | 0.3003 | 0.8225 | 0.7497 | 0.4837 |
| Proposed | **0.8379** | **0.7645** | 0.6698 | **0.8257** | **0.9025** | **0.5865** | **0.3646** | **0.8334** | **0.7652** | **0.5049** |

Figure 8 shows the semantic segmentation results of different methods on WHU-OPT-SAR. We also used four groups of images of complex scenes for comparison. In the first group, the intersecting roads, scattered water and villages are hard to classify. The results of DenseASPP, CCNet, MCANet and the proposed method are relatively good. DeepLabV3+ and DANet are not accurate enough to describe the scattered regions and boundaries between different classes. In the second group, since the characteristics of cities and villages are similar, the results of DenseASPP and MCANet are somehow affected, so parts of city regions are regarded as villages. The proposed method and CCNet achieved preferable segmentation. In the third group, the distribution of objects in the source image is more complex. In the forest, there are pairwise interactions among water, farmland and villages, and there are multiple intersecting roads. The roads and water are not well-represented by DeepLabV3+ and DANet. Other methods work well in this scenario. In the last group, the roads, cities and villages are densely distributed, and water crisscrosses farmland. The ground objects are reflected more accurately by MCANet and the proposed method. For TAFFN, objects like narrow roads, similar villages and cities and irregular water cannot be reflected well in multiple groups of images. We can see that the numerical results and the classification maps of this dataset are also matched.

After the experimental analysis on the two datasets, it can be seen that the proposed model behaves well both on numerical results and classification maps. Meandering water, scattered villages and intersecting roads are all well segmented. More importantly, there are rarely large areas of misclassification or missing objects. The method has strong adaptability in complex scenes with various ground objects. The proposed framework, feature extraction strategies and joint loss function do play a big role in classification. The ablation experiments are explained in detail in the next section.

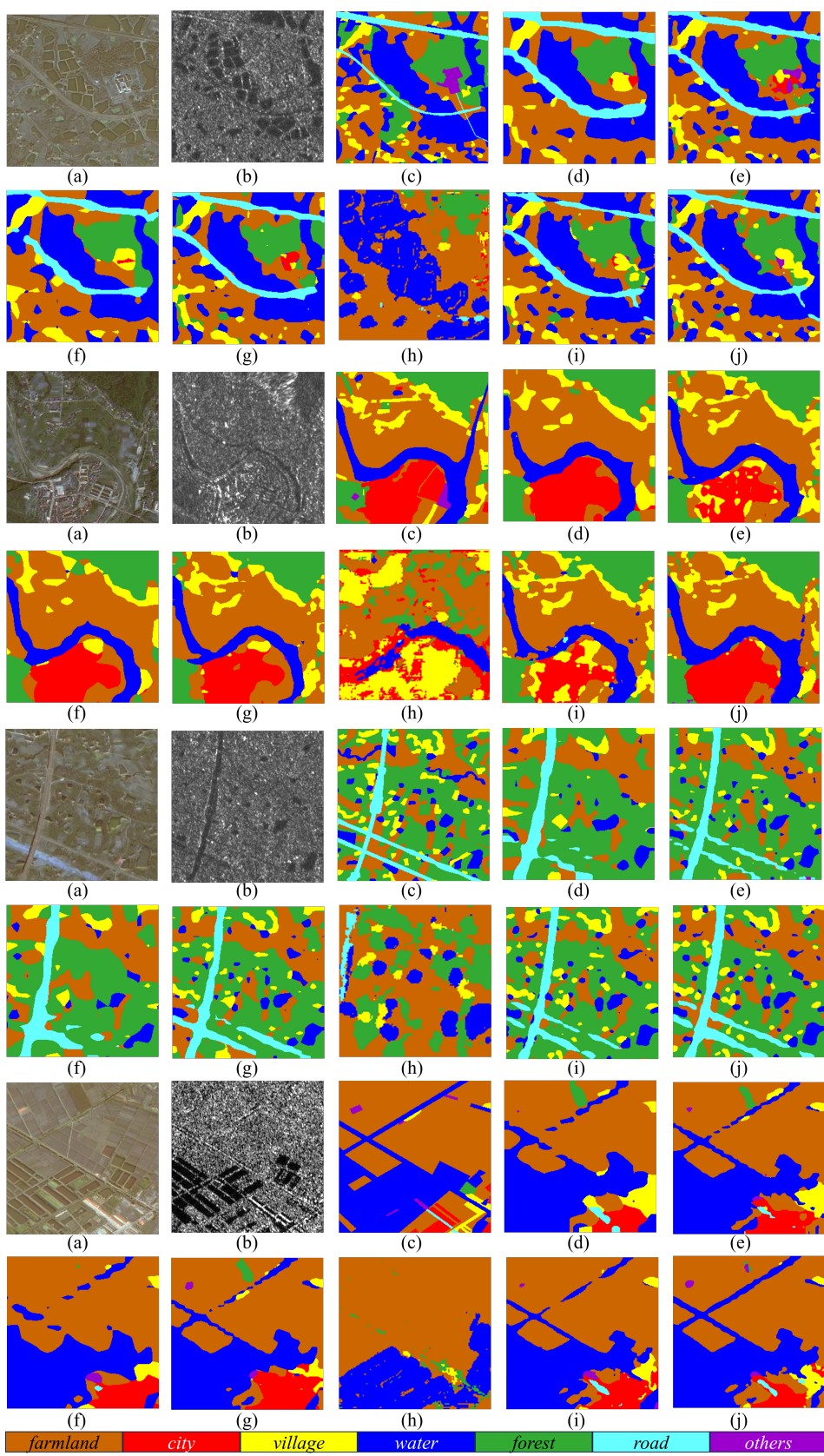

**Figure 8.** Land cover classification maps of different models on WHU-OPT-SAR. (**a**) Optical images. (**b**) SAR images. (**c**) Semantic labels. (**d**) DeepLabV3+. (**e**) DenseASPP. (**f**) DANet. (**g**) CCNet. (**h**) TAFFN (128 × 128). (**i**) MCANet. (**j**) Proposed SAMFNet.

## 4. Discussion

### 4.1. Computational Complexity

To further illustrate the efficiency of different methods, the number of parameters and the amount of computation are counted in Table 3. When training the network, growing the parameters increases the space complexity of the model. Devices with large video memory are needed. The amount of computation determines the execution time. It depends on the computing power of the GPU, so there are requirements for the flops of the hardware chip. We assume that for all models, the inputs are the coupled SAR and optical images with a size of $256 \times 256$. The values obtained under the two datasets are very close, since only the number of output channels of the last layer is different. For parameters, the proposed method is only inferior to TAFFN. There are no other pretrained backbones like ResNet or VGGNet added to the proposed model. It should be noted that although TAFFN has few parameters, its computational complexity is relatively high. Once the batch size increases, it is difficult for general GPUs to meet the requirements of flops. CCNet and MCANet performed well in the previous experimental analysis, but the number of parameters was indeed large. As to computation, although the flops of DeepLabV3+ and DANet are smaller, it is difficult for them to achieve desirable outcomes. The result of the proposed model is suboptimal. The computational complexity of CCNet and MCANet is relative high. In summary, the parameters and computational costs of the proposed method are low, and we can also obtain the best segmentation results. This will greatly improve the practicability of the model.

**Table 3.** Comparison of parameters and computation.

| Models | Params | FLOPs | Input Tensor |
|---|---|---|---|
| DeepLabV3+ | 39.05 M | 13.25G | |
| DenseASPP | 35.39 M | 39.40G | |
| DANet | 47.45 M | 14.41G | SAR: |
| CCNet | 70.95 M | 79.99G | [1,1,256,256] |
| TAFFN | 0.31 M | 33.13G | OPT: |
| MCANet | 85.93 M | 102.39G | [1,4,256,256] |
| Proposed | 19.60M | 28.78G | |

### 4.2. Analysis of Different Attention Mechanisms

In this subsection, CBAM [35], SP [41] and the designed SAEM are compared under the unified framework proposed in Figure 3. Only part of the attention mechanism is different. In CBAM, the attention module consists of channel attention and spatial attention. Conventional max pooling and average pooling are used for feature transformation. In the original SP, only strip average pooling is introduced to generate the long receptive fields. We added two more branches with strip max pooling to SP. Then, multi-dimensional feature superposition, fusion and conversion were implemented to obtain more contextual information. From the numerical results in Table 4, we can see that SP performed slightly better than CBAM. The proposed model with SAEM achieved the best results on the three metrics, which were basically 1% higher.

Figure 9 further illustrates the effect of different attention mechanisms through three groups of classification maps. In the first group, although all three models obtained good classification results, the proposed method can describe the distribution of grassland more accurately. In the second group, the water is the focus to be classified, and it is clearly delineated by the proposed method. In the last group, the scattered barren are clearly classified by the proposed method, while the other two models only focus on large area. It can be seen that the designed SAEM is conducive to salient feature extraction and improves the fineness of segmentation.

**Table 4.** Comparison of attention mechanisms.

| Models | OA | Kappa | mIoU |
|---|---|---|---|
| Att-CBAM | 0.8658 | 0.8363 | 0.6640 |
| Att-SP | 0.8680 | 0.8390 | 0.6681 |
| Proposed-SAEM | 0.8763 | 0.8492 | 0.6853 |

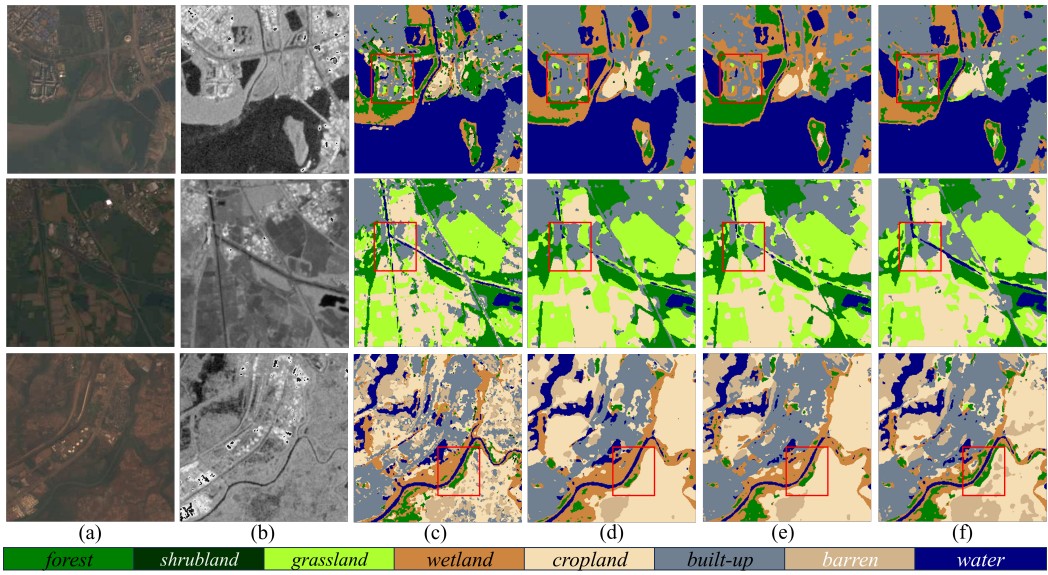

forest — shrubland — grassland — wetland — cropland — built-up — barren — water

**Figure 9.** Land cover classification maps on DFC2020. (**a**) Optical images. (**b**) SAR images. (**c**) Semantic labels. (**d**) Att-CBAM. (**e**) Att-SP. (**f**) Proposed-SAEM.

### 4.3. Analysis of Multi-Scale Feature Extraction

Multi-scale semantic information is very important for image segmentation. The designed MSEM combines the ASPP [38] and convolutions with different kernels (MSCK) to obtain high-level features. We want to verify the effect of the two parts. Three cases of contrast experiments were carried out. Modules with ASPP or MSCK and with both were compared. It should be noted that the number of feature maps outputted by the original MSEM module is 256 channels. Therefore, when implementing the above two experiments, the output of each part was also adjusted to 256 channels. In Table 5, the validation values of the first two experiments are close, but the model with joint modules is superior. This set of experiments proves that the combination of multi-scale information is helpful to improve the classification accuracy.

**Table 5.** Comparison of multi-scale modules.

| Cases | Multi-Scale Module | | OA | Kappa | mIoU |
|---|---|---|---|---|---|
| | **ASPP** | **MSCK** | | | |
| 1 | ✓ | - | 0.8734 | 0.8456 | 0.6784 |
| 2 | - | ✓ | 0.8724 | 0.8441 | 0.6739 |
| 3 | ✓ | ✓ | 0.8763 | 0.8492 | 0.6853 |

### 4.4. Analysis of the Joint Loss Function

The joint loss function used in this paper contains three loss terms. $L_{CE}$ is used as the basic loss to constrain and guide the training process. Therefore, it was retained in all experiments. $L_{Focal}$ is a commonly used loss function to solve class imbalance. $L_{SIoU}$ can further smoothly update the gradients, which, in turn, reduces oscillation during training. Then, there are four combinations for comparison ($L_{CE}$, $L_{CE} + L_{Focal}$, $L_{CE} + L_{SIoU}$ and $L_{CE} + L_{Focal} + L_{SIoU}$). Table 6 shows the results of models with different compositions of

losses. We can see that the model with only $L_{CE}$ can help to obtain good accuracy. The joint application of $L_{Focal}$ or $L_{SIoU}$ can effectively improve the segmentation effect, but only one of them is insufficient. The joint loss function with the three terms achieved the best results. In follow-up research, we are going to learn other losses or adjust the proportion of each loss to achieve better constraints.

**Table 6.** Comparison of loss functions.

| Cases | Loss Function | | | OA | Kappa | mIoU |
|---|---|---|---|---|---|---|
| | $L_{CE}$ | $L_{Focal}$ | $L_{SIoU}$ | | | |
| 1 | ✓ | - | - | 0.8691 | 0.8404 | 0.6730 |
| 2 | ✓ | ✓ | - | 0.8738 | 0.8460 | 0.6781 |
| 3 | ✓ | - | ✓ | 0.8736 | 0.8455 | 0.6758 |
| 4 | ✓ | ✓ | ✓ | 0.8763 | 0.8492 | 0.6853 |

### 4.5. Application in Other Multi-Modal Segmentation Tasks

In order to show the applicability of the proposed method, supplementary experiments were carried out on the MFNet dataset [49]. This RGB–thermal dataset consists of 1569 pairs of RGB and thermal images and has been applied in autonomous driving systems. It should be noted that the number of pixels of each class is unbalanced, and there is a large amount of unlabeled pixels. Since the number of classes and the channels of the multi-modal images are inconsistent with the previous datasets, the inputs and the size of confusion matrix need to be modified accordingly when executing the proposed model. Figure 10 shows the segmentation maps of four urban scenes. The first two were taken during daytime, and the others were taken at nighttime. The results of MFNet were generated by the demo code provided by the author. It can be seen that the proposed method can achieve better classification results. Both color cones and bumps can be clearly segmented. Meanwhile, numerical metrics were also compared. The class accuracy and mIoU of the proposed method reached 0.8144 and 0.6912, respectively, which are both higher than the optimal values of the original MFNet. The proposed model does have a certain application potential in other multi-modal segmentation tasks.

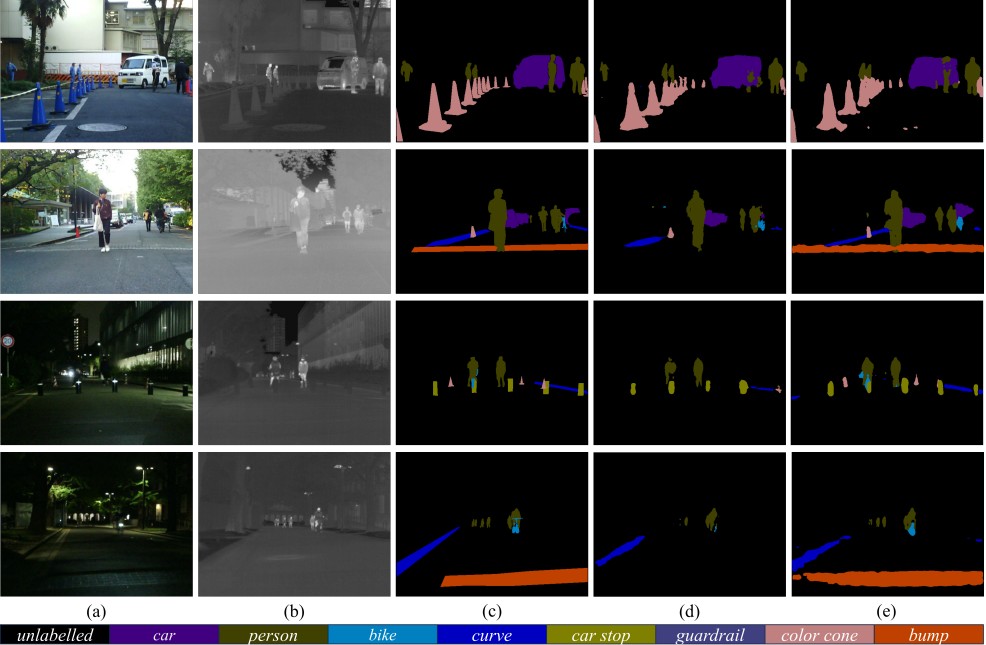

**Figure 10.** Urban scene segmentation maps on the RGB–thermal dataset. (**a**) RGB images. (**b**) Thermal images. (**c**) Semantic labels. (**d**) MFNet. (**e**) Proposed SAMFNet.

### 5. Conclusions

In this paper, we proposed a two-branch semantic segmentation network for land cover classification with multi-modal optical and SAR images. The numerical results and segmentation maps demonstrated various advantages of the proposed method. First, the novel symmetric attention mechanism with multiple long receptive fields can extract more contextual information. Objects with different shapes in the original images are perceived well. Secondly, multi-scale semantic fusion is implemented to enrich complementary information. High-level features extracted by dilated and varisized convolutions and low-level features from shallow layers are all considered and integrated together. Thirdly, a symmetrical structure and multiple plug-and-play modules were adopted to build the model. It has strong flexibility and adaptability. This was verified on an RGB–thermal dataset. Furthermore, the computational complexity of the proposed model is relatively low, and high classification accuracy was achieved. All these advantages prove the effectiveness of the method. However, the current research still depends heavily on the labeled dataset. In the future, we will deeply explore the implementation of semi-supervised and weakly supervised methods and study a lightweight network so that the model can be applied to more practical scenarios.

**Author Contributions:** Conceptualization, D.X.; methodology, D.X. and Z.L.; software, D.X. and Z.L.; validation, D.X., Z.L. and H.F.; formal analysis, D.X. and Z.L.; investigation, D.X., Z.L. and H.F.; resources, D.X.; data curation, D.X. and F.W.; writing—original draft preparation, D.X.; writing—review and editing, F.W., Z.L. and H.F.; visualization, D.X.; supervision, Y.W.; project administration, D.X.; funding acquisition, F.W. All authors have read and agreed to the published version of the manuscript.

**Funding:** This research was funded by the National Key R&D Program of China (No. 2022YFB3902300).

**Data Availability Statement:** The relevant methods, data and results of this study can be exchanged and shared in depth after communicating with the corresponding author.

**Acknowledgments:** The authors thank the 2020 IEEE GRSS Data Fusion Contest and Wuhan University for providing the aligned multi-modal data.

**Conflicts of Interest:** The authors declare no conflicts of interest.

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
