# Peer review of "Multi-Scale Feature Fusion Network with Symmetric Attention for Land Cover Classification Using SAR and Optical Images"

_remotesensing, doi:10.3390/rs16060957_

Round 1

Reviewer 1 Report

Comments and Suggestions for Authors

Manuscript ID: remotesensing-2842114

Title: Multi-scale Feature Fusion Network with Symmetric Attention for Land Cover Classification Using SAR and Optical Images

Recommendations for Authors:

1.              In the abstract, the authors must improve it and could add the manuscript's novelty and topicality.

2.              The authors should improve the introduction section, add mention the classification type used in order to mention the gaps in previous studies and the innovations in this study using recently published papers ( after 2020), and clarify the aims of this study.

3.              The authors should explain the classification method and the pre-processing of the data.

4.              In figure 3 : the authors should improve the resolution.  

5.              The authors should improve the discussion part in order to compare the results with previous work and show the novelty.

6.              The authors should improve the conclusion section and show the main results obtained and the novelty in this study.

Comments on the Quality of English Language

Moderate editing of English language required

Author Response

Dear reviewer,

On behalf of my co-authors, we thank you very much for giving us the constructive comments and suggestions on our manuscript entitled “Multi-scale Feature Fusion Network with Symmetric Attention for Land Cover Classification Using SAR and Optical Images”. We have tried our best to revise our manuscript according to the comments.

According to the comments of the reviewers, the changes made to this article mainly include the following aspects:

(1) We reorganize and edit the Abstract. The novelty of the proposed method and how it deals with the current difficulties are elaborated.

(2) The Introduction section has been improved. The classification type is described in more detail. The challenges of the current research as well as the innovations of this article are reedited.

(3) In the Results and Discussion sections, the results are reanalyzed. Compared with other methods, the characteristics and advantages of the proposed method are supplemented.

(4) The Conclusion section has also been modified. The main results and novelties of the article are summarized.

(5) Extensive editing of English language has been implemented. The manuscript is checked by several colleagues fluent in English writing. In order not to affect the reading fluency, minor changes and changes that do not affect the meaning are not marked.

The revised manuscript is presented in the mode of "Track Changes". The following contents are response to your constructive comments.

Thank you and best regards.
Sincerely yours,

Dongdong Xu

Comments and Suggestions:

Comments 1: In the abstract, the authors must improve it and could add the manuscript's novelty and topicality.

Response 1: Thank you very much for your suggestion. The previous summary is indeed too bland. We have reorganized and edited the abstract. The novelty of the proposed method and how it deals with the current difficulties are elaborated. (Changes in lines 1 to 15)

Comments 2: The authors should improve the introduction section, add mention the classification type used in order to mention the gaps in previous studies and the innovations in this study using recently published papers (after 2020), and clarify the aims of this study.

Response 2: Your suggestions are very valuable. The Introduction section has been improved in three aspects. Firstly, the deep learning-based methods are subdivided according to the fusion level. The most dominant feature-level methods are the research hotspots. Among them, the supervised segmentation models are focused. The classification type of the proposed model belongs to this category. Secondly, after analysis and comparison, several challenges in previous studies are summarized again. Lastly, we further clarify the aims and innovations of this study. (Changes in lines 56 to 77, lines 87 to 108, lines 116 to 139)

Comments 3: The authors should explain the classification method and the pre-processing of the data.

Response 3: Thank you very much for your suggestion. The classification method has been supplemented to the introduction section and the pre-processing of the data is added in subsection 2.1. (Changes in lines 56 to 77, lines 159 to 169)

Comments 4: In figure 3 : the authors should improve the resolution.

Response 4: Thank you very much for your suggestion. The resolution of Figure 3 has been effectively improved. At the same time, we also made adjustments to Figure 4, Figure 5 and Figure 6.

Comments 5: The authors should improve the discussion part in order to compare the results with previous work and show the novelty.

Response 5: Your suggestion is very constructive. In addition to the description of the figures, more analyses of the results are added in the Results and Discussion sections. Compared with other methods, the characteristics and advantages of the proposed method are supplemented. (Changes in lines 366 to 402, lines 441 to 460, lines 491 to 509, lines 513 to 518, lines 544 to 547)

Comments 6: The authors should improve the conclusion section and show the main results obtained and the novelty in this study.

Response 6: Your suggestion is very valuable. The Conclusion section has also been modified. The main results and novelties of the article are summarized. (Changes in lines 569 to 579)

Comments on the Quality of English Language: Moderate editing of English language required.

Response: Thank you very much for your suggestion. The manuscript is checked by several colleagues fluent in English writing. Extensive editing of English language has been implemented.

Reviewer 2 Report

Comments and Suggestions for Authors

In lines 75 to 89

There is no agreement about the challenges you have mentioned and there are many sources that have already given these answers to these challenges. More sources are needed for proof.

In lines 106 to 108

What is the reason for choosing these images? Are there no other images available?

 on page 5 and figure 1

A SAR image and four optics images have the same processing steps. How can it be explained?

Table 1, page 10

It shows that your proposed method has good results in all classes and for all methods. The question is, how are the classes that are compared with DFC2020 extracted?

In Table 1, the accuracy of the water class in the proposed method is 0.995. It seems that compared to image B, image C has visually performed better than the proposed method.

Author Response

Dear reviewer,

On behalf of my co-authors, we thank you very much for giving us the constructive comments and suggestions on our manuscript entitled “Multi-scale Feature Fusion Network with Symmetric Attention for Land Cover Classification Using SAR and Optical Images”. We have tried our best to revise our manuscript according to the comments.

According to the comments of the reviewers, the changes made to this article mainly include the following aspects:

(1) We reorganize and edit the Abstract. The novelty of the proposed method and how it deals with the current difficulties are elaborated.

(2) The Introduction section has been improved. The classification type is described in more detail. The challenges of the current research as well as the innovations of this article are reedited.

(3) In the Results and Discussion sections, the results are reanalyzed. Compared with other methods, the characteristics and advantages of the proposed method are supplemented.

(4) The Conclusion section has also been modified. The main results and novelties of the article are summarized.

(5) Extensive editing of English language has been implemented. The manuscript is checked by several colleagues fluent in English writing. In order not to affect the reading fluency, minor changes and changes that do not affect the meaning are not marked.

The revised manuscript is presented in the mode of "Track Changes". The following contents are response to your constructive comments.

Thank you and best regards.
Sincerely yours,

Dongdong Xu

Comments and Suggestions:

Comments 1: In lines 75 to 89-There is no agreement about the challenges you have mentioned and there are many sources that have already given these answers to these challenges. More sources are needed for proof.

Response 1: Thank you very much for your suggestion. The whole Introduction section has been improved in several aspects. Firstly, the deep learning-based methods are subdivided according to the fusion level. The most dominant feature-level methods are the research hotspots. Among them, the supervised segmentation models are focused. Secondly, after analysis and comparison, current challenges in previous studies are summarized again. Lastly, we further clarify the aims and innovations of this study. (Changes in lines 56 to 77, lines 87 to 108, lines 116 to 139)

Comments 2: In lines 106 to 108-What is the reason for choosing these images? Are there no other images available?

Response 2: Your questions are understandable. Currently, publicly available and registered multi-modal datasets for semantic segmentation with SAR and optical images are very scarce. In addition to the two datasets mentioned in this article, the datasets used in other articles include PoDelta dataset, Libourne dataset, and other self-built datasets from different regions. These datasets were searched at the beginning of our research, but we have yet to get them. If other datasets can be obtained in the future, we will conduct relevant validation and analysis.

Comments 3: on page 5 and figure 1-A SAR image and four optics images have the same processing steps. How can it be explained?

Response 3: Thank you for your careful review. We are sorry for the misunderstanding. The n1 and n4 in the figure actually represent the number of channels of the input SAR (one channel) and optical (four channels) images respectively. Each pair of inputs contains one SAR and one optical image. Both images are followed by convolution calculations, which convert inputs with different numbers of channels into 64-channel outputs. An explanation of the meaning of n is supplemented below Figure 3.

Comments 4: Table 1, page 10-It shows that your proposed method has good results in all classes and for all methods. The question is, how are the classes that are compared with DFC2020 extracted? In Table 1, the accuracy of the water class in the proposed method is 0.995. It seems that compared to image B, image C has visually performed better than the proposed method.

Response 4: Thank you for your careful review. We adopt a supervised approach for model learning, and different classes have been prelabeled in the label images. By calculating the confusion matrix between the pixel-level segmentation results outputted by the model and the labeled images, classification accuracy of each class can be obtained. As for the visual effect of water, image a, image b, and image c represent the input optical image, SAR image and the labeled reference image respectively. Image d to image j represent the segmentation results of different methods. Image c is a pre-annotated result and is the learning target of various methods, so image c looks better than all other images.

Reviewer 3 Report

Comments and Suggestions for Authors

The authors propose an end-to-end semantic segmentation network with symmetric attention and multi-scale fusion modules. The experimental results show that the proposed method achieves good results. Here are some comments:

1. The problem description in the abstract is not sufficiently concise and the methodology described in the abstract fails to make clear the problem that the methodology can solve.

2. For the description of the innovation points in Introduction: firstly, the model is described in innovation point 1 as being applicable to other multimodal segmentation and detection tasks with appropriate adjustments, has it been validated? What is the effect? It is suggested to add experimental validation of this expression; secondly, innovation point 2 is just a simple description of the method, which fails to logically reflect the role played by the method, and thus fails to effectively express the innovation; thirdly, innovation point 3 is not logical, may I ask what is the rationale for categorising the results as one innovation simply with the loss function and experimental effect?

3. The structure diagrams provided are vague and some words are more difficult to read.

4. In 3.3.1 'The MCANet obtains the suboptimal results since the mIoU of the 312 CCNet is slightly lower', may I ask if the two comparators in this sentence are reversed? Inconsistent with the results in Table 1.

5. In the RESULTS as well as the DISCUSSION section, there is a failure to analyse the results in depth, e.g. suggesting reasons why the method is better than others, suggesting reasons why the method performs better in certain categories, etc. Please think fully and add your own understanding rather than just describing the picture.

6. The description of the pictures of the experimental results did not manage to highlight the superiority of the method proposed in this paper.

7. Overall, the paper does not clearly analyse what problems can be solved by the proposed method, it is only a simple description of the proposed method and experiments, it does not analyse in depth where the method is superior, what role each module plays, and it lacks its own understanding of the proposed method.

Comments on the Quality of English Language

N/A.

Author Response

Dear reviewer,

On behalf of my co-authors, we thank you very much for giving us the constructive comments and suggestions on our manuscript entitled “Multi-scale Feature Fusion Network with Symmetric Attention for Land Cover Classification Using SAR and Optical Images”. We have tried our best to revise our manuscript according to the comments.

According to the comments of the reviewers, the changes made to this article mainly include the following aspects:

(1) We reorganize and edit the Abstract. The novelty of the proposed method and how it deals with the current difficulties are elaborated.

(2) The Introduction section has been improved. The classification type is described in more detail. The challenges of the current research as well as the innovations of this article are reedited.

(3) In the Results and Discussion sections, the results are reanalyzed. Compared with other methods, the characteristics and advantages of the proposed method are supplemented.

(4) The Conclusion section has also been modified. The main results and novelties of the article are summarized.

(5) Extensive editing of English language has been implemented. The manuscript is checked by several colleagues fluent in English writing. In order not to affect the reading fluency, minor changes and changes that do not affect the meaning are not marked.

The revised manuscript is presented in the mode of "Track Changes". The following contents are response to your constructive comments.

Thank you and best regards.
Sincerely yours,

Dongdong Xu

Comments and Suggestions:

The authors propose an end-to-end semantic segmentation network with symmetric attention and multi-scale fusion modules. The experimental results show that the proposed method achieves good results. Here are some comments:

Comments 1: The problem description in the abstract is not sufficiently concise and the methodology described in the abstract fails to make clear the problem that the methodology can solve.

Response 1: Thank you very much for your suggestion. The previous summary is indeed too bland. We have reorganized and edited the abstract. The novelty of the proposed method and how it deals with the current difficulties are elaborated. (Changes in lines 1 to 15)

Comments 2: For the description of the innovation points in Introduction: firstly, the model is described in innovation point 1 as being applicable to other multimodal segmentation and detection tasks with appropriate adjustments , has it been validated? What is the effect? It is suggested to add experimental validation of this expression; secondly, innovation point 2 is just a simple description of the method, which fails to logically reflect the role played by the method, and thus fails to effectively express the innovation; thirdly, innovation point 3 is not logical, may I ask what is the rationale for categorising the results as one innovation simply with the loss function and experimental effect?

Response 2: Your suggestions are very valuable. The whole Introduction section has been improved, and we further clarify the innovations of this study. In innovation point 1, the overall framework of the proposed model is described emphatically. In innovation point 2, the novel modules and their roles are explained. In innovation point 3, we focus on the better results, strong applicability and high efficiency of the proposed method. Experimental validation has been added on another RGB-Thermal segmentation task. The source images of the dataset are three channel visible images and one channel infrared images. Since the number of classes and the channels of the multi-modal images are inconsistent with the previous datasets, the inputs and the size of confusion matrix need to be modified accordingly when executing the proposed model. The details are described in subsection 4.5. (Changes in lines 116 to 139, lines 553 to 567)

Comments 3: The structure diagrams provided are vague and some words are more difficult to read.

Response 3: Thank you for your careful review. The resolutions of Figure 3, Figure 4, Figure 5 and Figure 6 have been effectively improved. At the same time, the words in the figures are resized accordingly.

Comments 4: In 3.3.1 'The MCANet obtains the suboptimal results since the mIoU of the 312 CCNet is slightly lower', may I ask if the two comparators in this sentence are reversed? Inconsistent with the results in Table 1.

Response 4: Thank you for your careful review. The description in this sentence is inaccurate. Among the three metrics, CCNet has two higher values than MCANet, and only the mIoU is a bit lower. It is not appropriate to directly draw this conclusion. The relevant description in the article has been revised. The comparison between the methods is reanalyzed. (Changes in lines 366 to 395)

Comments 5: In the RESULTS as well as the DISCUSSION section, there is a failure to analyse the results in depth, e.g. suggesting reasons why the method is better than others, suggesting reasons why the method performs better in certain categories, etc. Please think fully and add your own understanding rather than just describing the picture.

Response 5: Your suggestions are very constructive. In addition to the description of the figures, more specific analysis of the results is added in the Results and Discussion sections. The characteristics and advantages of the proposed method are supplemented. (Changes in lines 366 to 402, lines 441 to 460, lines 491 to 509, lines 513 to 518, lines 544 to 547)

Comments 6: The description of the pictures of the experimental results did not manage to highlight the superiority of the method proposed in this paper.

Response 6: Your suggestion is very valuable. We did not do a good job in analyzing the experimental results. The reasons behind the results have not been thoroughly considered. In the modified manuscript, we will try our best to explain the experimental results and highlight the superiority of the proposed method. (Changes in lines 366 to 402, lines 441 to 460, lines 491 to 509, lines 513 to 518, lines 544 to 547)

Comments 7: Overall, the paper does not clearly analyse what problems can be solved by the proposed method, it is only a simple description of the proposed method and experiments, it does not analyse in depth where the method is superior, what role each module plays, and it lacks its own understanding of the proposed method.

Response 7: Thank you very much for your pertinent suggestions. After receiving your review comments, we have re-examined this study. The gaps in previous research and the innovations in the proposed method have been further clarified. The Abstract and the Introduction section are improved, so that the novelty and topicality of the method can be well reflected. Then the framework and the novel modules are explained more clearly. In the Results and Discussion sections, more understandings of the method are added for analysis. Furthermore, the main results and novelties of the article are re-summarized in the Conclusion section. Thank you again for your guidances and suggestions.

Comments on the Quality of English Language: Extensive editing of English language required.

Response: Thank you very much for your suggestion. The manuscript is checked by several colleagues fluent in English writing. Extensive editing of English language has been implemented. The main changes are as follows:

  • Some long sentences have been modified into short sentences and unnecessary clauses have been removed.
  • The non-standard use of nouns to modify nouns has been reduced and corrected.
  • Excess phrases and suffixes are removed from the sentences.

Round 2

Reviewer 3 Report

Comments and Suggestions for Authors

Thanks for the revision.

I recommend for publication.

Comments on the Quality of English Language

None.